Methods

# Optimized ChIP-seq method facilitates transcription factor profiling in human tumors

Abhishek A Singh[1,2,*], Karianne Schuurman[1,*], Ekaterina Nevedomskaya[1,2], Suzan Stelloo[1], Simon Linder[1], Marjolein Droog[1], Yongsoo Kim[1], Joyce Sanders[3], Henk van der Poel[4], Andries M Bergman[5,6], Lodewyk FA Wessels[2,7], Wilbert Zwart[1,8]

Chromatin immunoprecipitation (ChIP)-seq analyses of transcription factors in clinical specimens are challenging due to the technical limitations and low quantities of starting material, often resulting in low enrichments and poor signal-to-noise ratio. Here, we present an optimized protocol for transcription factor ChIP-seq analyses in human tissue, yielding an ~100% success rate for all transcription factors analyzed. As proof of concept and to illustrate general applicability of the approach, human tissue from the breast, prostate, and endometrial cancers were analyzed. In addition to standard formaldehyde fixation, disuccinimidyl glutarate was included in the procedure, greatly increasing data quality. To illustrate the sensitivity of the optimized protocol, we provide high-quality ChIP-seq data for three independent factors (AR, FOXA1, and H3K27ac) from a single core needle prostate cancer biopsy specimen. In summary, double-cross-linking strongly improved transcription factor ChIP-seq quality on human tumor samples, further facilitating and enhancing translational research on limited amounts of tissue.

## Introduction

Steroid hormone receptors are not only critical regulators in human physiology, but also central players in multiple diseases, including cancer. The steroid hormone receptor family is composed of multiple members, including estrogen receptor α (ERα), estrogen receptor β, androgen receptor (AR), glucocorticoid receptor, progesterone receptor, and mineralocorticoid receptor. Around 75% of all human breast tumors are positive for, and considered dependent on, ERα. ERα is a hormone-dependent transcription factor, which upon activation by its natural ligand estradiol, binds regulatory regions throughout the genome to orchestrate responsive gene activity by chromatin looping (Fullwood et al, 2009; Flach & Zwart, 2016). This mode of activation is shared by practically all steroid hormone receptors, including AR. AR is considered the oncogenic driver in prostate cancer development and progression (Lonergan & Tindall, 2011). Both ERα and AR require direct functional involvement of pioneer factors, such as Forkhead box protein A1 (FOXA1), to facilitate chromatin accessibility at designated binding sites for ERα and AR (Robinson & Carroll, 2012).

Many transcription factors, such as steroid hormone receptors and FOXA1 (Swinstead et al, 2016), are intrinsically dynamic when it comes to chromatin interactions. ERα chromatin interactions are stabilized upon ligand binding, a step crucial for estradiol-mediated gene transcription and breast cancer cell proliferation (Tan et al, 2011). Analogous to this, AR in prostate cancer cells is confined in the cytosol prior to testosterone binding (Brinkmann et al, 1999). When activated, AR translocates into the nucleus to facilitate chromatin binding and testosterone-driven transcription. But even when activated, ERα (Swinstead et al, 2016) as well as AR (Kang et al, 2002) binds the chromatin in a dynamic and transient fashion, which is in contrast to the stable histone modifications that make up a structural and stable factor of the chromatin.

To date, most reports on hormone receptor genomics, including ERα and AR, made use of cell line models, and only slowly, reports on genome-wide chromatin binding in the context of human tumor tissue are being released. In breast cancer, we (Jansen et al, 2013; Severson et al, 2018) and others (Ross-Innes et al, 2012) identified distinct subsets of ERα chromatin binding profiles with prognostic potential, enabling the stratification of patients on outcome. Comparable observations were reported in prostate cancer, in which AR chromatin binding signatures were indicative of patient outcome

[1]Divisions of Oncogenomics, Oncode Institute, Netherlands Cancer Institute, Amsterdam, the Netherlands  [2]Molecular Carcinogenesis, Oncode Institute, Netherlands Cancer Institute, Amsterdam, the Netherlands  [3]Department of Pathology, Netherlands Cancer Institute, Amsterdam, the Netherlands  [4]Department of Urology, Netherlands Cancer Institute, Amsterdam, the Netherlands  [5]Division of Oncogenomics, Netherlands Cancer Institute, Amsterdam, the Netherlands  [6]Division of Medical Oncology, Netherlands Cancer Institute, Amsterdam, the Netherlands  [7]Faculty of Electrical Engineering, Mathematics, and Computer Science, Delft University of Technology, Delft, the Netherlands  [8]Laboratory of Chemical Biology and Institute for Complex Molecular Systems, Department of Biomedical Engineering, Eindhoven University of Technology, Eindhoven, the Netherlands

Correspondence: w.zwart@nki.nl
*Abhishek A Singh and Karianne Schuurman contributed equally to this work

(Sharma et al, 2013; Stelloo et al, 2015). However, due to technical limitations in working with human tumor tissue, most reports were typically limited to large quantities of starting material derived from surgery.

Over the recent years, many laboratories have developed novel technologies to further optimize Chromatin immunoprecipitation (ChIP)-seq analyses for low quantities of cells, including the addition of carriers (Zwart et al, 2013) and improved sequencing library preparation using transposase-mediated sequence indexing (Marine et al, 2011). More recently, a micro-fluidics/barcoding technology has been reported to enable ChIP-seq analyses for histone modifications on a single-cell level (Rotem et al, 2015). These and other technological advances have greatly improved immunoprecipitation efficiency and found intelligent ways to resolve a "minimal quantity of cells" problem. Still, when applying these approaches in clinical specimens, other challenges exist, which currently prevent the generation of high-quality ChIP-seq profiles for transcription factors, including sub-optimal tissue fixation. Recently, the use of additional fixatives next to standard formaldehyde (FA) has been reported to increase ChIP efficiency in cell line models (Tian et al, 2012; Engelen et al, 2015; Puc et al, 2015) and *Drosophila* embryos (Aoki et al, 2014). The application of this novel approach in human tissue specimens remains unexplored.

Here, we report an improved ChIP-seq procedure, which involves implementation of disuccinimidyl glutarate (DSG) as an additional fixative next to standard FA fixation, greatly enhancing the quality of hormone receptor ChIP-seq analyses in human tumor tissue. Since data quality for histone modifications was not affected, the altered procedure exclusively increased the data quality for factors transiently interacting with the chromatin, including ERα and AR. With improved protocols for tissue ChIP-seq data analyses, results generated in cell lines and tissue may be better compared, further facilitating translational research in hormone-dependent cancers.

## Materials and Methods

### Cell culture and clinical specimens

The luminal breast cancer cell line MCF-7 and prostate cancer cell line LNCaP were cultured in DMEM and RPMI-1640, respectively (Gibco, Thermo Fisher Scientific), supplemented with 10% fetal bovine serum (Sigma-Aldrich). Fresh frozen tumor samples were obtained from postoperative tumor tissue at the Netherlands Cancer Institute (Amsterdam, the Netherlands) from the patients who received no neoadjuvant endocrine treatment. Tumor content was assessed by haematoxylin and eosin (H&E) stainings on slides taken throughout the tissue sample, and only samples comprised of 65% tumor or more were processed. For further information on tumor samples, see Table S1.

### Sample cross-linking and immunoprecipitations

ChIP was performed as described previously (14) with some adaptations. DSG (20593; Thermo Fisher Scientific) was dissolved in dry DMSO to a concentration of 0.5 M. This stock was aliquoted and stored at –20°C, aliquots were defrosted and used directly, and excess was discarded.

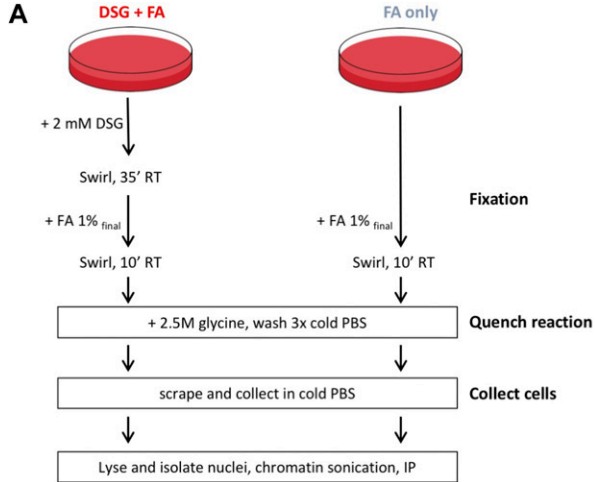

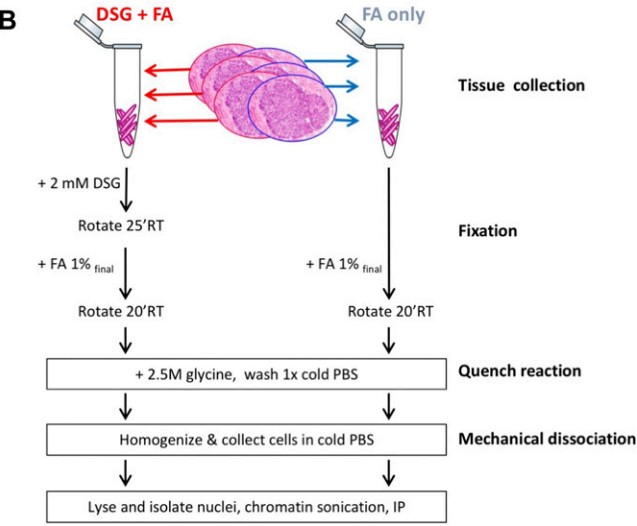

**Figure 1.  Experimental design.**
Overview of experimental design for cell lines **(A)** and human tumor tissue **(B)**.

For cell line experiments, ~25–30 million cells were used per fixation method, per experiment. For cells fixed with FA only, FA was added to the medium to 1% final concentration, plates were swirled and incubated for 10 min at room temperature, followed by the addition of a surplus of 2.5 M glycine (1/20 of total volume) to quench the reaction. For cells fixed with DSG+FA, the medium was aspirated and the cells were covered with either solution A (MCF-7, 50 mM Hepes-KOH, 100 mM NaCl, 1 mM EDTA, 0.5 mM EGTA) or PBS (LNCaP, PBS + 1 mM $MgCl_2$, 1 mM $CaCl_2$) containing 2 mM DSG for 35 min at room temperature, after which FA was added to 1% final, and cells were incubated for another 10 min at room temperature. The reaction was quenched by adding a surplus of glycine. After quenching, sample processing was identical for DSG+FA and FA only samples. In short, plates were washed with cold PBS three times, and cells were scraped and collected in cold PBS. Nuclei were isolated and the chromatin was sheared using a PicoBioruptor (Diagenode). Chromatin shearing was confirmed by agarose gel

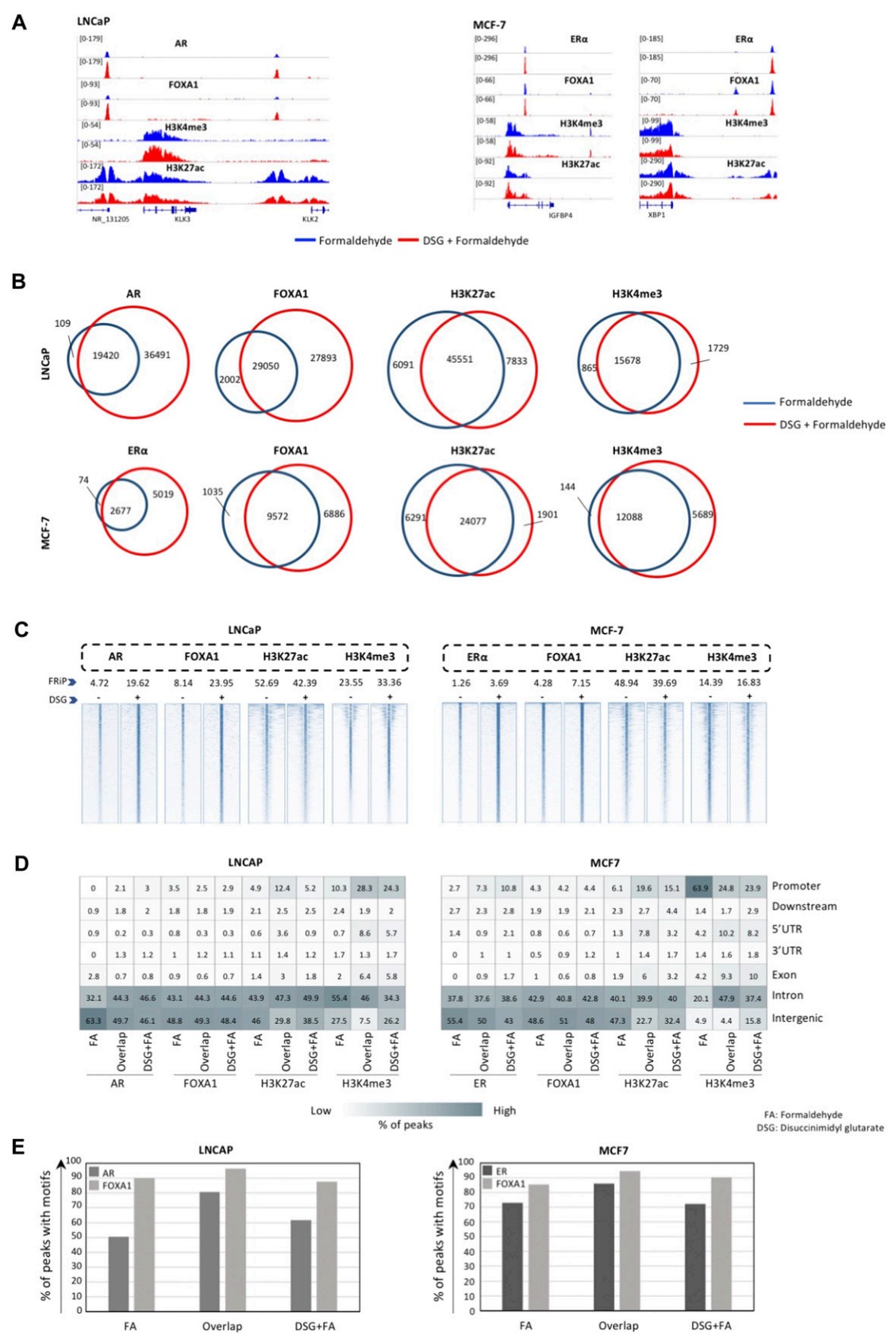

electrophoresis (cell lines) and bioanalyser (tissue samples), which appeared not substantially affected by the fixation method that was used (Fig S1A and B). After immunoprecipitation, 10 RIPA washes were performed, followed by one TBS wash and reverse cross-linking. DNA was isolated as described (14).

Fresh frozen tissue was cryosectioned into 30-µm-thick slices, collected in Eppendorf tubes and stored at –80°C till processing. Number of sections used per sample and surface area for each tissue sample are shown in Table S1. During cryosectioning, alternating slices were collected for fixation by DSG+FA or FA only, respectively, to prevent differences in samples due to tissue heterogeneity. Samples fixed with DSG+FA or FA only were processed in parallel. For FA only, tissue sections were fixed in solution A containing 1% FA for 20 min at room temperature while rotating, followed by the addition of glycine for quenching. For DSG+FA, fixation was started by adding solution A containing 2 mM DSG to the tissue sections and rotating for 25 min at room temperature, followed by the addition of FA to 1% final concentration and another 20-min incubation at room temperature. The reaction was quenched by adding a surplus of glycine. Further sample processing was identical for DSG+FA and FA only samples. In short, after quenching tissue sections were pelleted and washed with cold PBS. A motorized pellet pestle (Sigma-Aldrich) was used to disrupt the tissue in cold PBS and obtain a cell suspension, after which the nuclei were isolated and the chromatin was sheared. During immunoprecipitation, human control RNA (4307281; Thermo Fisher Scientific) and recombinant Histone 2B (M2505S; New England Biolabs) were added as carriers.

For both cell line and tissue ChIPs, 5 µg of antibody and 50 µl of magnetic protein A or G beads (10008D or 10009D; Thermo Fisher Scientific) were used per IP. The following antibodies were used: ERα (HC-20, sc-543 Santa Cruz Biotechnology lot K1715 and F1215), Foxa1/2 (M-20, sc-6554 Santa Cruz Biotechnology lot G1415 and D1015), AR (N-20, sc-816 Santa Cruz Biotechnology lot C0916 and I0215), H3K27ac (39133Active Motif lot 31814008), and H3K4me3 (ab8580 Abcam lot GR240214-2). For primer sequences used in ChIP-quantitative PCR (QPCR) validation experiments, see Table S2.

## Solexa sequencing, data alignment, and peak calling

Immunoprecipitated DNA was processed for sequencing using standard protocols and sequenced on an Illumina Hi-seq 2500 with 65-bp single end reads. Sequenced samples were aligned to the reference human genome (Ensembl release 55: Homosapiens GRCh 37.55) using Burrows-Wheeler Aligner (Li and Durbin 2009, 2010), reads with a mapping quality >20 were further used (Li et al, 2009). Peak calling was performed using MACS2 (Zhang et al, 2008) with option -nomodel and Dfilter (v1.6) (Kumar et al, 2013) with options -bs = 50 -ks = 20 -refine -nonzero for nuclear receptors, -bs = 100

-ks = 60 for H3K27ac, and -bs = 100 -ks = 100 for H3K4me3 mark, where only peaks were considered that were shared by the two peak callers.

## Bioinformatics

Genome browser snapshots were generated using Integrative Genomics Viewer (Thorvaldsdottir et al, 2013). The intensity plots and fraction of reads in peaks score (FriP) were generated using scripts from package deepTools2 (Ramirez et al, 2016). The Galaxy Cistrome (Liu et al, 2011) server was used to compute genomic distribution of peaks (Shin et al, 2009) and distribution of motifs around the binding sites along with the percentage of peaks with the motifs (Wang et al, 2013).

# Results

## DSG improves cross-linking in cell lines of transcription factors

Chromatin immunoprecipitation is a standard procedure to study transcription factor/chromatin interactions. Even though FA is classically used to cross-link protein/DNA complexes to enable immunoprecipitation of DNA regions of interest, other fixatives are available. Previously, a "two-step" double-cross-linking method was reported, in which sequentially protein–protein interactions were fixed using DSG, followed by a protein–DNA cross-linking (FA) (Tian et al, 2012). As this approach was reported to significantly increase ChIP efficiency, we applied this procedure to the most frequently used model systems in breast and prostate cancer research: MCF-7 and LNCaP cells, respectively. Proliferating cells were used in these analyses and fixed either using standard 1% FA or through a two-step fixation procedure with 45-min 2-mM DSG fixation, of which the last 10 min the cells are co-incubated with 1% FA (see the Materials and Methods section, Fig 1A). Between fixation methods, no clear effect was seen on DNA fragment size distribution (Fig S1A). For AR (in LNCaP) and ERα (in MCF-7), clear peaks were observed proximal to their classic target genes in the corresponding cell lines: KLK3 for AR while IGFBP4 and XBP1 for ERα. As expected, these peaks were shared with FOXA1 and histone modification H3K27ac, while this histone modification was also found at promoter regions along with H3K4me3 (Fig 2A). ChIP signal for ERα, AR, and FOXA1 was substantially increased when cells were fixed using DSG, both on a single-locus scale (Fig 2A) and genome-wide (Fig 2B and C). These results were confirmed by ChIP-QPCR (Fig S2A and B), where clear enrichment over IgG-negative control was observed for all ChIP samples analysed. For all transcription factors, consistent higher enrichment was found in samples fixed with DSG and FA as compared to FA alone, while results for both histone modifications were more variable. In concordance with this, the

**Figure 2. DSG increases TF ChIP-seq quality in cell line models.**
**(A)** Left: Genome browser snapshots of AR, FOXA1, H3K4me3, and H3K27ac binding in LNCAP cells (left) with and without DSG treatment (FA: blue. FA+DSG: red). Right: Genome browser snapshots of ERα, FOXA1, H3K4me3, and H3K27ac in MCF7 cells (right) with and without DSG (FA: blue. FA+DSG: red). Tag count and gene IDs are indicated. **(B)** Venn diagram showing the overlap of binding sites using FA (blue) and FA along with DSG (red) in LNCAP (top) and MCF7 (bottom) cells for AR, ERα, FOXA1, H3K4me3, and H3K27ac. **(C)** Intensity plots showing the read densities over the peaks (±5 kb) in profiles generated using FA (−) and FA along with DSG (+). The fraction of reads in peaks (FRiP score) is placed above each intensity plot. **(D)** Genomic distribution for sites shared or unique to FA with or without DSG, using LNCAP and MCF7 cells. **(E)** Comparison of percentage of binding sites with AR and FOXA1 motifs in LNCAP cells as well as for ERα and FOXA1 motifs in MCF7 cells for sites using FA with or without DSG.

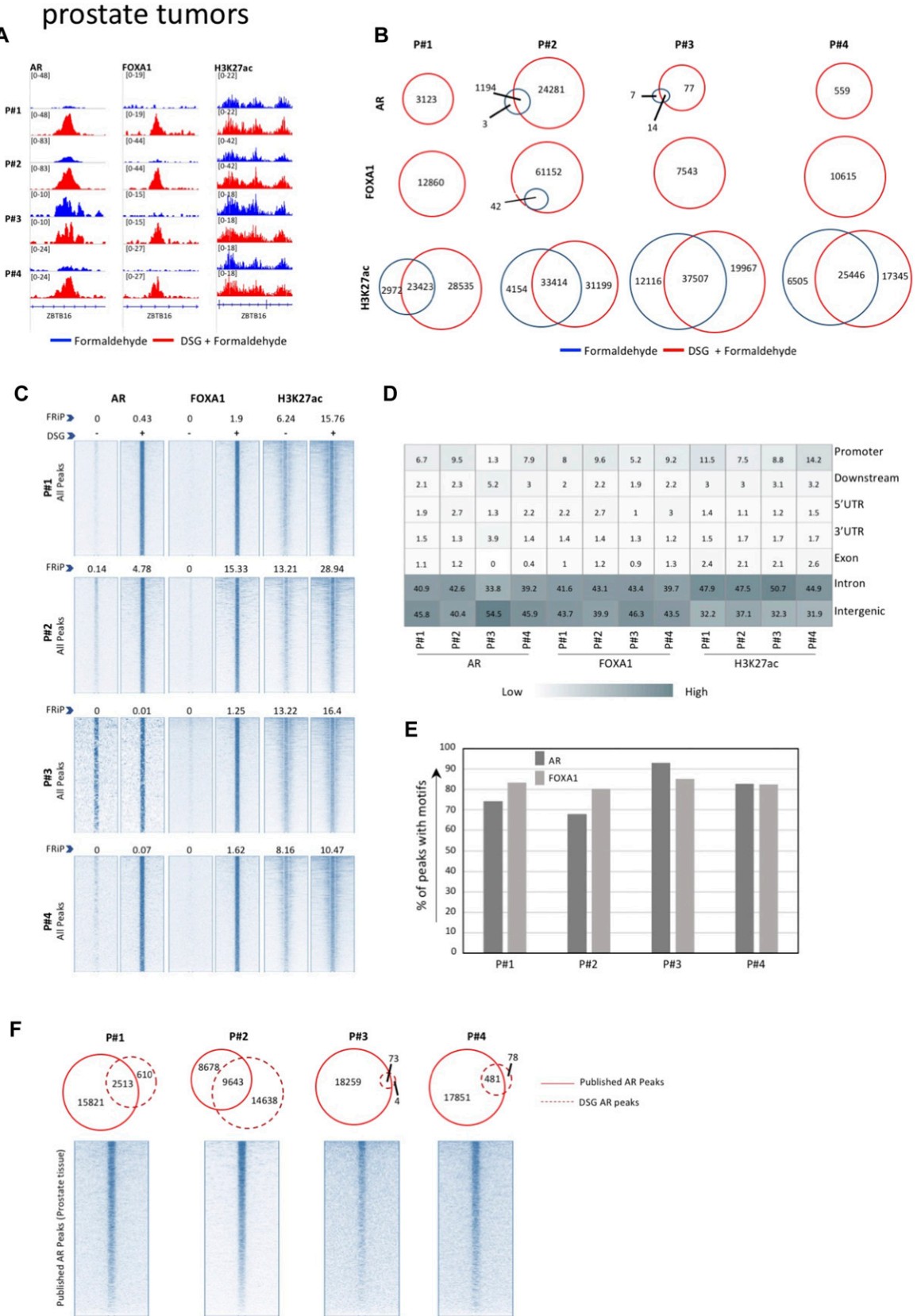

Fraction of Reads in Peaks (FRiP score) was clearly increased for ERα, AR, and FOXA1 (Fig 2C). Importantly, the increased ChIP-seq signal was uniformly found on a genome-wide scale (Fig S3B). No clear bias in peak distribution is observed for specific genomic regions (Fig 2D), even though an increased percentage of promoter-enriched peaks was found for both ERα and AR, which is possibly related to the low number of peaks analysed for these subsets. The relative orientation to the most proximal canonical motif remained largely unaffected (Figs 2E and S3A). For the studied histone modifications H3K4me3 and H3K27ac in both cell lines, the vast majority of peaks were shared between the two fixation methods (Fig 2A–D). However, still regions were found selectively enriched for cells only fixed with FA versus those fixed with FA and DSG. Therefore, these findings were further confirmed using ChIP-QPCR. For some of the "DSG only" or "FA only" peaks, enrichment was found using this method (Fig S4). These data suggest potential false-negative peaks with the currently applied peak calling algorithms, as we have reported previously (Zwart et al, 2016).

## DSG improves cross-linking in tumor specimens of transcription factors

By double fixation using DSG and formaldehyde, we found a substantial increase in chromatin binding signal and number of binding sites for ERα (MCF-7), AR (LNCaP), and FOXA1 (MCF-7 and LNCaP) without affecting H3K4me3 and H3K27ac (Fig 2). To explore if this fixation would also improve data quality for ChIP-seq in clinical specimens, we collected fresh frozen primary tumor specimens from four ERα-positive breast tumors and four prostate tumors, from our hospital biobank. To illustrate the generalizability of the procedure, we also included three ERα-positive endometrial tumors in our analyses. All tissues were cryosectioned and collected in pre-chilled tubes in an alternating fashion, ruling out tissue heterogeneity as a potential confounding factor in our analyses (Fig 1B). Subsequently, tissue was processed in parallel, by either (1) standard 20 min fixation using 1% FA or (2) fixation using 2 mM DSG for 45 min, with the last 20 min in the presence of 1% FA. Samples were processed fully in parallel, using standard procedures which included mRNA/Histone 2B carriers (see the Materials and Methods section) (Zwart et al, 2013). Using bioanalyser analyses, no differences were observed in DNA fragment size distribution in any of the tumor types analysed, as exemplified for three tumor samples (Fig S1B).

For all three tumor types, a substantial increase in data quality was observed for ERα, AR, and FOXA1 with strong increased peak intensity (Figs 3A, 4A, and 5A), as compared to single fixation procedure using FA. These results were validated by ChIP-QPCR analysis of all factors analysed (prostate tumors: Fig S5, breast tumors: Fig S6, endometrial tumors: Fig S7), in which consistently

enrichment over a negative control region and IgG control was observed.

Also, on a genome-wide scale, a strong increase in signal was observed for all transcription factors studied in prostate cancer (Fig 3B and C), breast cancer (Fig 4B and C), and endometrial cancer (Fig 5B and C). The addition of DSG to the standard procedure significantly increased the enrichment of the factors, and this was reflected in the detection of greater number of binding sites. In accordance with this, higher FRiP score was observed for all transcription factors, in all three tumor types (Figs 3C, 4C, and 5C). The signal increase was uniformly distributed over all chromatin binding sites studied, implying no bias in genomic selectivity of the increased binding (Figs S8B, S9E, and S10B). Further, the distribution of the peaks relative to the most proximal gene was comparable between tumors (Figs 3D, 4D, and 5D) and cell lines (Fig 2D), which was in agreement with the previously published profiles for these factors (Jansen et al, 2013; Severson et al, 2018). As expected, the motif for the transcription factor analysed (ERα, AR, and FOXA1) was found enriched in 50–80% of the peaks (Figs 3E, 4E, 5E, S8A, S9D, and S10A), which is again in line with the cell line-based results (Fig 2E).

Importantly, for practically all samples studied, FA fixation alone was insufficient to yield any detectable signal with the low tissue quantity we used, and double fixation uniformly and substantially increased signal quality for all transcription factors tested. With a large quantity of breast tissue available, we expanded our analyses to H3K4me3, comparing both fixation methods. The results were in line with H3K27ac results in cell lines and tumors, i.e., no consistent significant differences were observed in terms of number of peaks detected (Fig S9A), signal enrichment (Fig S9B), distribution of the peaks relative to the most proximal gene (Fig S9C), and signal distribution over all chromatin binding sites studied (Fig S9E). However, in contrast to cell line data, for tumor samples a substantial number of histone modification peaks were preferentially identified in either FA or DSG fixed samples. To address this, we re-analysed the H3K27ac ChIP-seq data from prostate (Fig 3B) breast (Fig 4B) and endometrial (Fig 5B) tumors. Based on raw read counts of the H3K27ac ChIP-seq data, substantial coverage was identified at all the peak subsets (shared between both fixation methods, DSG+FA unique, FA unique) for prostate (Fig S11), breast (Fig S12) and endometrial (Fig S13) tumors, for both fixation methods. Also, when separately analyzing the peaks that were identified in the individual tumors, coverage at these regions in all other samples was found, indicating strong overlap of peaks between samples. These data were further confirmed using ChIP-QPCR (Fig S14) and externally validated using publically available H3K27ac ChIP-seq data that were generated on breast (Patten et al, 2018) (Fig S15), prostate (Kron et al, 2017) (Fig S16), and endometrial (Droog et al, 2017) (Fig S17) cancer specimens. Cumulatively, these data suggest that rather than truly unique peaks, differential regions identified with the two

**Figure 3. ChIP-seq analyses in primary in prostate tissue.**
**(A)** Genome browser snapshot of AR, FOXA1, and H3K27ac binding profiles in primary prostate specimens, with and without DSG treatment (FA: blue. FA+DSG: red). Tag count and gene IDs are indicated. **(B)** Venn diagram showing the overlap of binding sites between the ChIP-seq profiles generated using FA (blue) and FA along with DSG (red) for AR, FOXA1, and H3K27ac. **(C)** Intensity plots showing the read densities over the peaks (±5 kb) in profiles generated using FA (−) and FA along with DSG (+). The FRiP score is placed above each intensity plot. **(D)** Genomic distribution of sites identified in prostate specimens fixed with FA and DSG (FA+DSG). **(E)** Percentage of sites containing AR and FOXA1 motifs, under FA+DSG conditions. **(F)** Venn diagram depicting union of all AR binding sites previously reported in primary prostate cancer versus the sites identified from samples fixed with FA and DSG. The intensity plot shows the read density of the previously published AR binding sites (±5 kb).

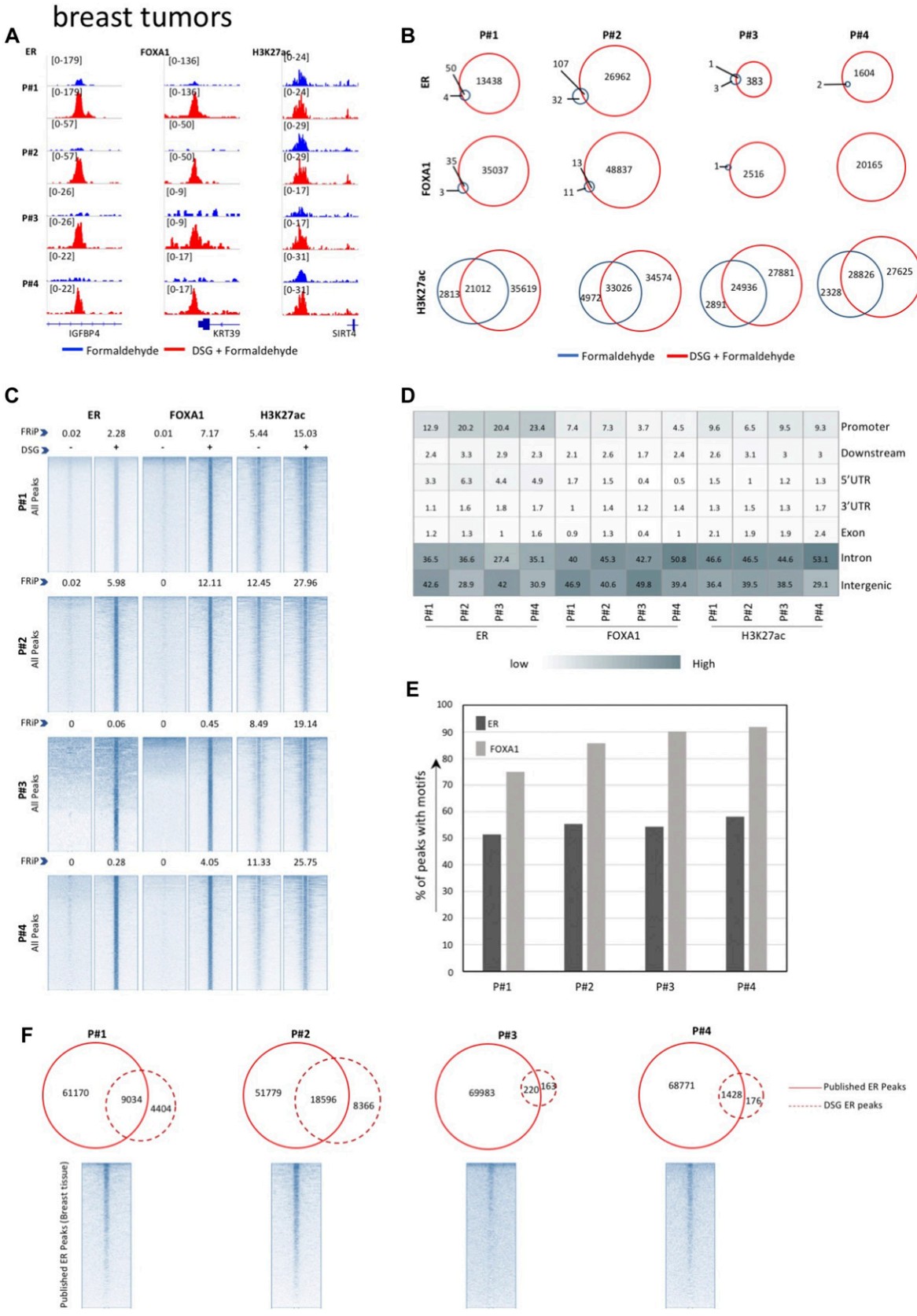

fixation methods are likely false negatives that arise due to the peak calling algorithms that were used, as we observed previously (Zwart et al, 2016).

Previous work reported ERα cistromics in primary breast cancer specimens (Ross-Innes et al, 2012). In accordance with this, we previously reported genome-wide AR chromatin binding in prostate cancers (Stelloo et al, 2015) and ERα profiles in endometrial cancers (Droog et al, 2017). We used all the peaks that were reported by the above-mentioned studies to generate a superset of all sites for ERα and AR ever found in the corresponding tumor type. Next, these sites were tested for overlap with the peaks we generated using DSG (Figs 3F, 4F, and 5F). In all cases, an overlap was found with the previously reported sites, confirming our results in other tumor ChIP-seq datasets.

### Double fixation enables high-quality ChIP-seq in 18G core needle biopsies

We observed that double fixation yields a clear increase in the number of transcription factor binding sites and improves signal intensity in all three tumor types tested. This would suggest that incorporation of DSG in the standard protocol also permits the use of smaller quantities of tissue material—often a requirement when using precious human surgical specimens. We gained access to two 18G core needle biopsies (diameter: 1.2 mm) of primary prostate cancers, which were cryosectioned and subsequently fixed with DSG/FA. For each tumor, we performed ChIP-seq for AR, FOXA1, and H3K27ac, yielding thousands of peaks even with this little quantity of input material (Fig 6A, genome-wide Fig 6B and C). Further, distribution of the peaks relative to the most proximal gene was in line with our observation in the LNCaP cell line (Fig 6D). Around 70% of AR and FOXA1 peaks identified were positive for AR and FOXA1 motifs, respectively (Figs 6E and S18), which is in the same order of magnitude as we identified in large prostate resection samples (Fig 3E) and cell line-based observations (Fig 2E).

Furthermore, we observed a strong overlap with a previously reported AR prostate peak set (Stelloo et al, 2015), suggesting limited false negativity of expected regions (Fig 6F). In summary, a modified ChIP-seq protocol with double fixation of DSG prior to FA enables genome-wide assessment of transiently binding transcription factors and H3K27ac in small 18G core needle biopsy samples with generation of high-quality data.

## Discussion

In studying hormone-dependent cancers, model systems are limited. As a direct consequence, the vast majority of studies are based on a relatively low number of cell line models and it becomes increasingly relevant to study steroid hormone receptor genomics in the context of human tissue specimens.

Gaining access to human tumor tissue is typically challenging, and tissue availability for translational research is typically limited. The development of early detection-screening programs, involving mammography screening for breast cancer, resulted in the detection and removal of tumors that are often small. Furthermore, due to limited clinical benefit of surgery on metastatic disease, tissue availability at this stage of the disease is low. With this, there is a strong need in translational research to further improve protocols that enable transcription factor profiling in very low amounts of human tumor specimens.

Over the recent years, multiple technological advances have greatly increased the sensitivity of ChIP-seq analyses for small sample numbers. These include the addition of carriers which diminish background and improve chromatin immunoprecipitation efficiency (Zwart et al, 2013). Also, sequencing library preparation procedures have improved significantly by incorporating tagmentation for adaptor annealing (Picelli et al, 2014). Improvements in sample barcoding in conjunction with microfluidics resulted in the development of ultra-sensitive protocols to enable single-cell ChIP-seq analyses on histone modifications (Rotem et al, 2015). Yet, since histones are an intrinsic part of the chromatin, proteins that dynamically interact with the DNA such as transcription factors are captured less easily.

Protein–DNA cross-linking in ChIP-seq experiments is typically performed using FA. However, proteins with rapid dynamics and transient chromatin interactions are inefficiently cross-linked, which directly affects ChIP efficiency (Schmiedeberg et al, 2009). Furthermore, as FA is a short-range cross-linker, protein–protein interactions are not very well stabilized using this method. As the transcription factor complex is composed of a large number of proteins, both for ERα (D'Santos et al, 2015) and for AR (Paltoglou et al, 2017; Stelloo et al, 2018), stabilizing the entire complex would potentially also stabilize DNA interactions thereof throughout the immunoprecipitation procedure.

We present an optimized protocol to enhance the transcription factor profiling in human tumor tissue and successfully performed ChIP-seq for ERα, AR, and FOXA1 in primary breast, endometrium, and prostate cancer specimens. Importantly, incorporation of DSG as an additional cross-linker and a sole variable sufficed to provide high-quality ChIP-seq data for samples, which would hardly yield any peaks using the standard protocol. This update in the protocol not only greatly increased the success-rate of ChIP-seq experiments when using precious human samples, but also enabled the use of small 18G core needle biopsies for ChIP-seq analyses on multiple factors.

With this, we present an easily implementable modification of the standard ChIP-seq procedure to substantially increase both

---

**Figure 4.   ChIP-seq analyses in primary breast tissue.**
**(A)** ChIP-seq overview of ERα, FOXA1, and H3K27ac binding profiles with and without DSG treatment (FA: blue. FA+DSG: red). Tag counts and gene IDs are indicated. **(B)** Venn diagram showing the overlap of binding sites between the ChIP-seq profiles generated using conventional cross-linker FA (blue) and FA along with DSG (red) for ERα, FOXA1, and H3K27ac. **(C)** Intensity plots showing the read densities over the peaks (±5 kb) in profiles generated using FA (−) and FA along with DSG (+). The FRiP score is placed above each intensity plot. **(D)** Genomic distribution of ERα and FOXA1 sites in samples processed with FA and DSG (FA+DSG). **(E)** Percentage of ERα and FOXA1 sites in FA/DSG fixed samples positive for ESR1 and FOXA1 motifs. **(F)** Venn diagram of union of ERα binding sites from published datasets and ERα sites identified in FA/DSG-treated sample. The intensity plot shows the read density of the previously published ERα binding sites (±5 kb).

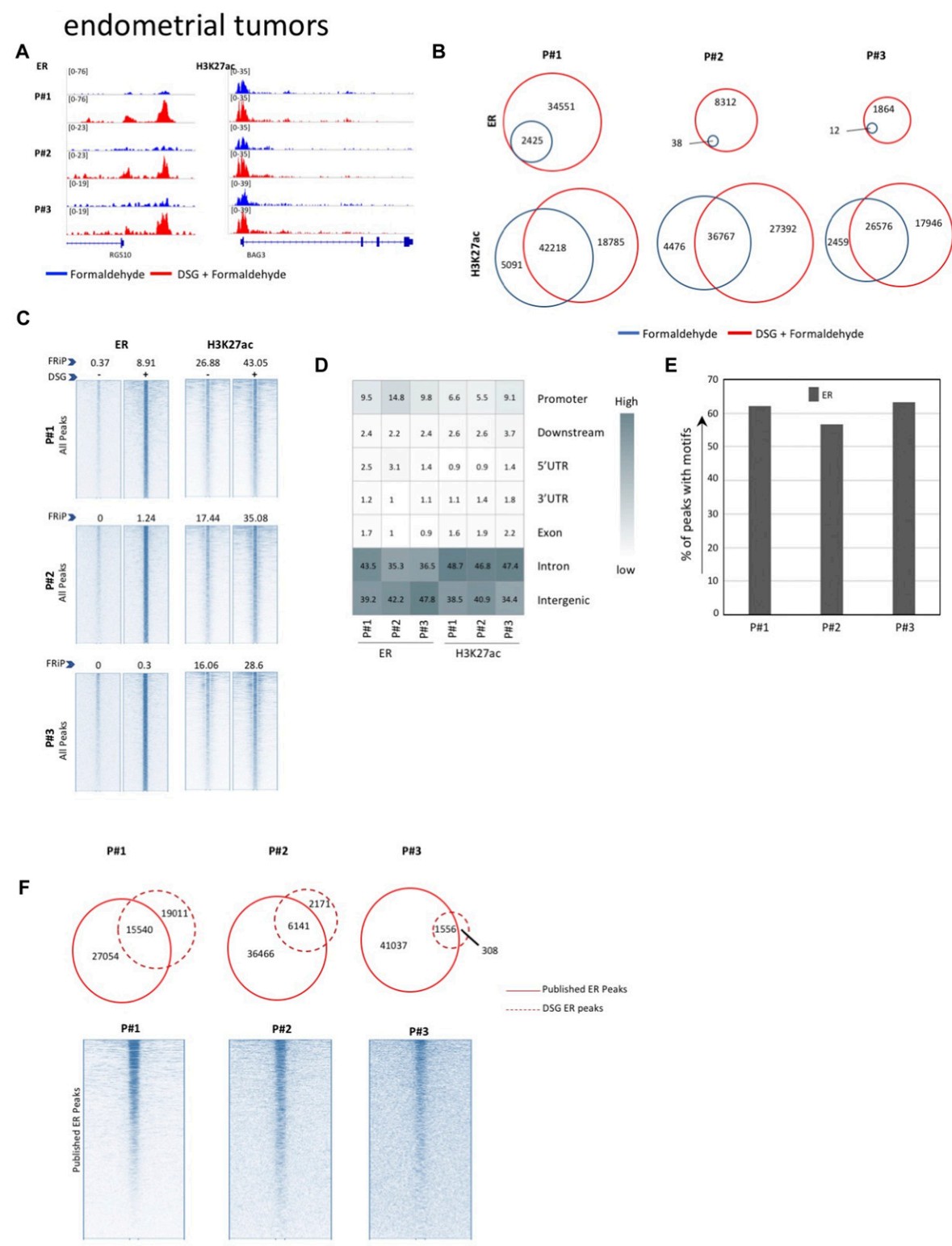

**Figure 5. ChIP-seq analyses in primary endometrial tissue.**
**(A)** Genome browser snapshots of ERα and H3K27ac sites in endometrial tumors, in FA-fixed samples with and without DSG treatment (FA: blue. FA+DSG: red).
**(B)** Venn diagram showing the overlap of sites for ERα and H3K27ac using FA (blue) and FA along with DSG (red). **(C)** Intensity plots showing the read densities over the peaks (±5 kb) in profiles generated using FA (−) and FA along with DSG (+). The fraction of reads in peaks (FRiP score) is placed above each intensity plot. **(D)** Genomic distribution of sites for ERα and H3K27ac in FA/DSG fixed samples (FA+DSG). **(E)** Percentage sites of ERα sites in FA/DSG fixed samples positive for ESR1 motifs.
**(F)** Venn diagram of union of all ERα sites previously reported in endometrial tumors and sites identified in FA/DSG-fixed samples. The intensity plot shows the read density of the previously published ERα binding sites (±5 kb).

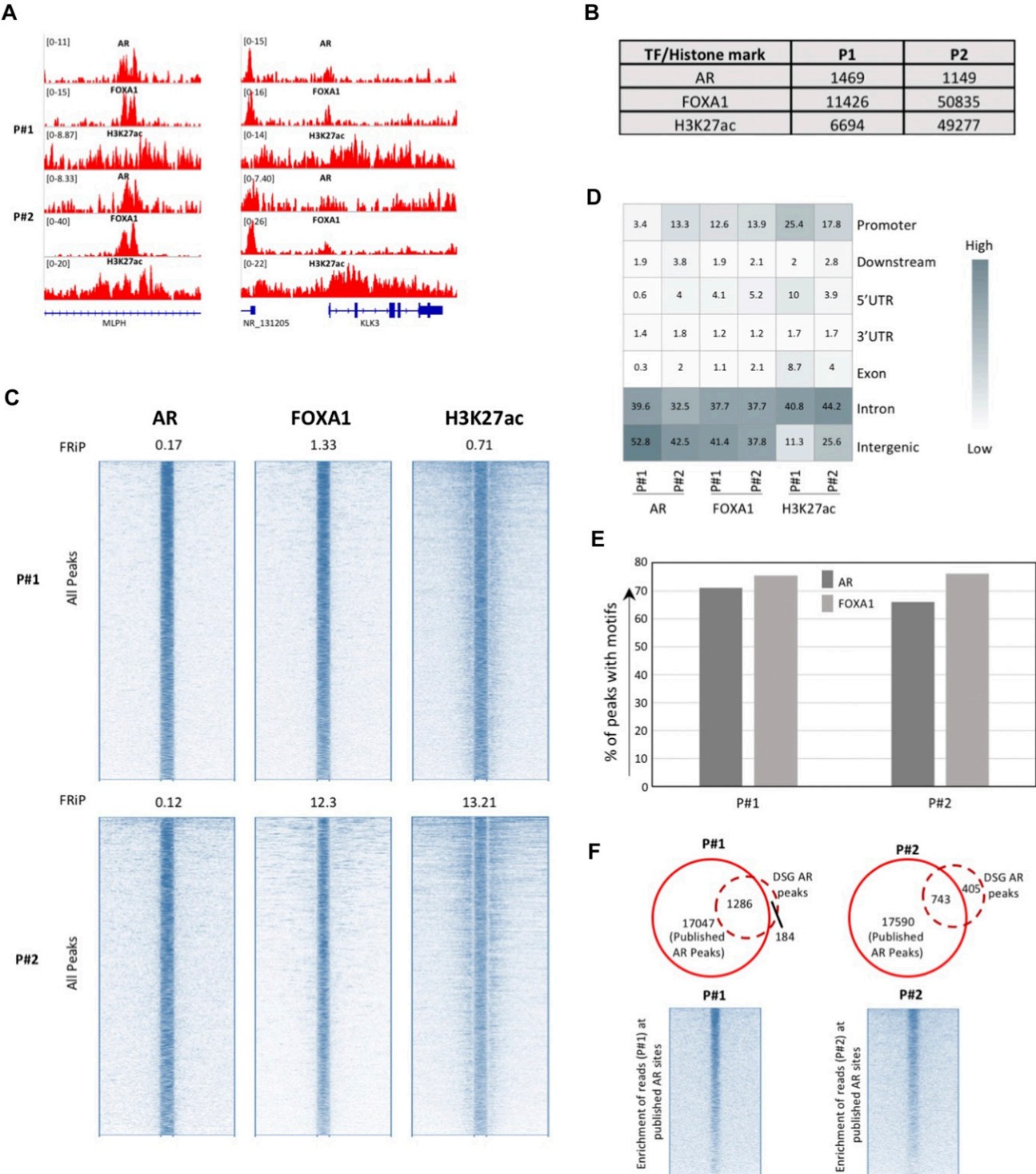

**Figure 6. Generating ChIP-seq profiles from 18G core needle biopsies from radical prostatectomy samples.**
**(A)** Genome browser snapshots for AR, FOXA1, and H3K27ac ChIP-seq with and without DSG treatment (FA: blue. FA+DSG: red). Tag count and gene IDs are indicated. **(B)** Number of binding sites for AR, FOXA1, and H3K27ac. **(C)** Intensity plots showing the read densities over the peaks (±5 kb) in samples fixed with FA (−) or FA along with DSG (+). The FRiP score is placed above each intensity plot. **(D)** Genomic distribution of AR, FOXA1, and H3K27ac binding sites. **(E)** Percentage of AR and FOXA1 binding sites that are positive of motifs for AR and FOXA1. **(F)** Venn diagram of AR (FA+DSG) binding sites and union of all AR sites previously reported in primary prostate cancers. The intensity plot shows the read density for the previously published AR binding sites (±5 kb).

data quality and quantity when studying transcription factor chromatin interactions in hormone-dependent cancers, facilitating translational research of hormone receptor cistromics in human samples.

## Data Availability

All data are available through the National Center for Biotechnology Information Gene Expression Omnibus under accession number GSE114737.

## Supplementary Information

## Acknowledgements

The authors would like to thank the NKI Core Facility Molecular Pathology and Biobanking for technical support and providing tissue specimens. This project is funded by the Dutch Cancer Society KWF. W Zwart is supported by a KWF/Alpe d'HuZes Bas Mulder Award and Dutch Scientific Organization NWO VIDI.

### Author Contributions

AA Singh: conceptualization, software, formal analysis, visualization, and writing—original draft, review, and editing.
K Schuurman: formal analysis, validation, methodology, and writing—original draft, review, and editing.
E Nevedomskaya: conceptualization, software, investigation, visualization, and writing—review and editing.
S Stelloo: data curation, methodology, and writing—original draft, review, and editing.
S Linder: data curation, formal analysis, validation, visualization, and writing—review and editing.
M Droog: data curation, formal analysis, and writing—review and editing.
Y Kim: software, visualization, and writing—review and editing.
J Sanders: data curation, formal analysis, and writing—review and editing.
H van der Poel: resources, supervision, and writing—review and editing.
AM Bergman: conceptualization, supervision, funding acquisition, project administration, and writing—review and editing.
LFA Wessels: resources, supervision, project administration, and writing—review and editing.
W Zwart: conceptualization, supervision, funding acquisition, methodology, project administration, and writing—original draft, review, and editing.

### Conflict of Interest Statement

The authors declare that they have no conflict of interest.

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
