## [Reviewer comments · Life Science Alliance]

Life Science Alliance

Optimized ChIP-seq procedure facilitates transcription factor profiling in human tumors.

Abhishek Singh, Karianne Schuurman, Ekaterina Nevedomskaya, Suzan Stelloo, Simon Linder, Marjolein Droog, Yongsoo Kim, Joyce Sanders, Henk van der Poel, Andre Bergman, Lodewyk Wessels, and Wilbert Zwart

DOI: [10.26508/lsa.201800115](https://doi.org/10.26508/lsa.201800115)

Corresponding author(s): Wilbert Zwart, Netherlands Cancer Institute

Review Timeline:

Submission Date:	2018-06-25
Editorial Decision:	2018-08-08
Revision Received:	2018-11-22
Editorial Decision:	2018-12-18
Revision Received:	2018-12-18
Accepted:	2018-12-18

Scientific Editor: Andrea Leibfried

Transaction Report:

August 8, 2018

Re: Life Science Alliance manuscript #LSA-2018-00115-T

Prof. Wilbert Zwart
Netherlands Cancer Institute
dept of Oncogenomics
PLEsmanlaan 121
Amsterdam, NH 1066CX
Netherlands

Dear Dr. Zwart,

Thank you for submitting your manuscript entitled "Optimized ChIP-seq procedure facilitates transcription factor profiling in human tumors." to Life Science Alliance. The manuscript was assessed by expert reviewers, whose comments are appended to this letter.

As you will see, the reviewers appreciate your optimised ChIP-seq method and support publication of a revised version of the manuscript here. Specifically, the data need to be confirmed with an alternative method (reviewer #2) and inconsistencies need to get addressed (both reviewers). Furthermore, the revised manuscript needs to provide more detailed information and discussion.

The requested changes all seem straightforward to address and we would thus like to ask you to provide a revised version addressing all concerns raised.

- A letter addressing the reviewers' comments point by point.
- An editable version of the final text (.DOC or .DOCX) is needed for copyediting (no PDFs).
- High-resolution figure, supplementary figure and video files uploaded as individual files: See our detailed guidelines for preparing your production-ready images, <http://life-science-alliance.org/authorguide>
- Summary blurb (enter in submission system): A short text summarizing in a single sentence the study (max. 200 characters including spaces). This text is used in conjunction with the titles of

papers, hence should be informative and complementary to the title and running title. It should describe the context and significance of the findings for a general readership; it should be written in the present tense and refer to the work in the third person. Author names should not be mentioned.

B. MANUSCRIPT ORGANIZATION AND FORMATTING:

Full guidelines are available on our Instructions for Authors page, <http://life-science-alliance.org/authorguide>

Thank you for this interesting contribution to Life Science Alliance. We are looking forward to receiving your revised manuscript.

Sincerely,

Reviewer #2 (Comments to the Authors (Required)):

In the manuscript entitled "Optimized ChIP-seq method facilitates transcription factor profiling in human tumors" Singh and colleagues present a method to perform ChIP-seq for transcription

factors in human tissue, based on a double crosslinking technique with DSG and formaldehyde. The manuscript is very well written and presented and therefore easy to follow, this optimized protocol will be highly relevant for the scientific community. However, confirmations of the method need to be provided in order to control for the specificity of the protocol. These will be necessary for publication in LSA.

Major comments

- 1- Generally, confirmations of the ChIP-seq data by ChIP qPCR, with IgG controls, need to be performed. This will ensure that the observed observations can be validated by an alternative method and that the observed results are specific.
- 2- Figures 3B, 4B, 5B and S3A: the authors report that, regarding histone modifications, no consistent significant differences were observed in term of number of peaks, signal enrichment, distribution relative to the most proximal gene and signal distribution over binding sites. However, the observed overlaps between Formaldehyde and DSG + Formaldehyde are not great (which they do not comment on). Compared to the cell lines, there is a big variability in which peaks are detected between the two fixation methods. The differences need to be addressed: why is there such a difference in overlap? Which ones are the true peaks? Would different fixation methods allow the detection of different peaks? Confirmations of several peaks for the ones detected in Formaldehyde only, DSG + Formaldehyde only, and the overlap are necessary.

Minor comments

- 1- Two types of font are used in the abstract.
- 2- Introduction second line: typo in diseases (a missing).
- 3- Page 6 line 6 typo in Histone (e missing).
- 4- Fig 6C: H3K27ac label missing above the figure.

Reviewer #3 (Comments to the Authors (Required)):

The paper entitled "Optimized ChIP-seq method facilitates transcription factor profiling in human tumors" by Singh et al. describes a technical improvement to increase ChIP-seq signal with particular application to transcription factors in frozen tissues. Improvements to the quality of ChIP-seq data from clinical samples and obtaining more data from limited amounts of clinical tissue are both important areas where advances are highly desired. The authors provide data from two cell lines and numerous clinical samples across three tumor types to demonstrate their methods superior performance as compared to standard ChIP-seq methods. Overall, they demonstrate a clear performance improvement over standard methods, however, additional information describing their experiments needs to be added to the manuscript.

- (1) In the section titled "DSG improves cross-linking in cell lines of transcription factors" the authors need to report the number of cells used for the ChIP-seq experiments.
- (2) In the section titled "DSG improves cross-linking in tumor specimens of transcription factors" the authors discuss the method's applicability to profile TFs from reduced amount of frozen tissue, but they don't discuss the amount of starting tissue used for the AR/ER, FOXA1 and H3K27ac profiles shown in figures 3 and 4 or indeed any of the tissue samples. The authors should detail the starting number of sections and an estimate of the cross-section size. Also, if available, the authors should report the amount of DNA in the soluble chromatin that was used for the immuno-precipitations.
- (3) The authors state towards the bottom of page 7 "... and no bias in peak distribution is observed for specific genomic regions (Figure 2D)." However Figure 2D shows that the percentage of

promoter peaks changes from 0% in FA to 3% in DSG+FA for AR. For ER the changes are 2.7% in FA to 10.8% in DSG+FA (a 4-fold increase). This seems to indicate a bias for promoter peaks in DSG+FA data that does not significantly detract from the method but should be acknowledged and discussed.

(4) In Figure 3 the authors show ChIP-seq analyses comparing AR, FOXA1 and H3K27ac in primary in prostate tissue with similar analyses in Figures 4 and 5. It is striking that virtually all the TF data using FA appears to have completely failed (many have zero peaks and/or zero FrIP scores). This is surprising since there is significantly better quality data that has been published from these tumor types for these TFs by multiple groups (as the authors reference in their paper). There needs to be an explanation for this apparent discordance with the literature results.

(5) In several places in the manuscript the authors state that there is no effect of double fixation on histone mark ChIP-seq (e.g. near the top of page 9: "The results were in line with H3K27ac results in cell lines and tumors, i.e. no consistent significant differences were observed in terms of number of peaks detected.") However, Figure 4B and 5B show consistent, large increases in the number of H3K27ac peaks identified with DSG+FA compared to FA. This increase should be acknowledged and there should be some effort to show these additional peaks are likely real (overlap with H3K27ac in cell line data, sequence conservation, etc.)

(6) Lastly, some considerations should be made regarding chromatin fragmentation after extensive fixation. It is expected that there will be an increased proportion of high DNA sizes in the soluble chromatin after the DSG/FA protocol compared to FA alone. The high sizes will reduce the amount of DNA that can be sequenced after the IP. The authors should discuss this point and show results for the fragment size distribution comparing both DSG/FA and standard FA.

Reviewer #2 (Comments to the Authors (Required)):

In the manuscript entitled "Optimized ChIP-seq method facilitates transcription factor profiling in human tumors" Singh and colleagues present a method to perform ChIP-seq for transcription factors in human tissue, based on a double crosslinking technique with DSG and formaldehyde. The manuscript is very well written and presented and therefore easy to follow, this optimized protocol will be highly relevant for the scientific community. However, confirmations of the method need to be provided in order to control for the specificity of the protocol. These will be necessary for publication in LSA.

We would like to thank the reviewer for the constructive feedback and valuable suggestions, and we are delighted to see the reviewer finds the manuscript well-written, and the protocol of high relevance for the community. As recommended by the reviewer, we now confirmed the method using other assays, and included additional controls. With this, we believe we have addressed the concerns raised by the reviewer, and we sincerely hope you agree.

Major comments

1- Generally, confirmations of the ChIP-seq data by ChIP qPCR, with IgG controls, need to be performed. This will ensure that the observed observations can be validated by an alternative method and that the observed results are specific.

As recommended, ChIP-QPCR confirmations as well as IgG controls have now been incorporated for all datasets, and now presented in the supplementary figures (S2, S4, S5, S6, S7, S14). All these data confirm our previous observations, and we thank the reviewer for this valuable suggestion.

2- Figures 3B, 4B, 5B and S3A: the authors report that, regarding histone modifications, no consistent significant differences were observed in term of number of peaks, signal enrichment, distribution relative to the most proximal gene and signal distribution over binding sites. However, the observed overlaps between Formaldehyde and DSG + Formaldehyde are not great (which they do not comment on). Compared to the cell lines, there is a big variability in which peaks are detected between the two fixation methods. The differences need to be addressed: why is there such a difference in overlap? Which ones are the trues peaks? Would different fixation methods allow the detection of different peaks? Confirmations of several peaks for the ones detected in Formaldehyde only, DSG + Formaldehyde only, and the overlap are necessary.

We thank the reviewer for drawing this to our attention. The reviewer is absolutely right: the overlap for the histone marks between formaldehyde and

DSG + formaldehyde is not great based on called peaks, and a statement on this is now clearly mentioned in the results section. To address the question of which peaks are the true peaks, and whether the different fixation methods would allow detection of different peaks, we have performed the following analyses:

- 1. For all peak subsets in Figures 3B, 4B, 5B and S3A, we now incorporated the raw data visualization coverage plots in Supplementary Figures S11, S12, S13. For each of the three peaksets (shared, formaldehyde-unique, DSG+ formaldehyde-unique), signal was in fact observed for all of those locations**
- 2. For H3K27ac ChIP-seq in tumor samples, other independent datasets have been published in endometrial cancer (Droog et al., 2017 PNAS), prostate cancer (Kron et al 2017 Nature Genetics) and breast cancer (Patten et al., 2018 Nature Medicine). For each of the H3K27ac peak subsets we identified for any of the tumors, we tested whether read coverage was observed in the publicly available datasets (Supplementary Figures S15, S16, S17). For each of the peak subsets (shared, formaldehyde-unique, DSG+ formaldehyde-unique), clear coverage was found in all the other datasets tested. Please note that consistently the 'shared between formaldehyde and DSG+formaldehyde' regions consistently showed the highest coverage for all samples tested.**
- 3. The histone modification ChIP-seq data were now further confirmed using ChIP-QCPR (Supplementary Figure S14). Again, for each of the subsets of peaks, enrichment was observed for all samples tested, which was consistently enriched over IgG negative control.**

Cumulatively, these controls show that rather than unique peaks, these 'DSG + formaldehyde' and 'formaldehyde alone' peaks should be considered as false-negative regions that were not picked up with the peak calling algorithm that was used. This is now explicitly mentioned in the results section, and all analyses mentioned are now included in the supplementary figures.

Minor comments

- 1- Two types of font are used in the abstract.
- 2- Introduction second lane: typo in diseases (a missing).
- 3- Page 6 lane 6 typo in Histone (e missing).
- 4- Fig 6C: H3K27ac label missing above the figure.

Thank you for the detailed corrections and feedback, which have now all been implemented in the revised version of the manuscript.

Reviewer #3 (Comments to the Authors (Required)):

The paper entitled "Optimized ChIP-seq method facilitates transcription factor profiling in human tumors" by Singh et al. describes a technical improvement to increase ChIP-seq signal with particular application to transcription factors in frozen tissues. Improvements to the quality of ChIP-seq data from clinical samples and obtaining more data from limited amounts of clinical tissue are both important areas where advances are highly desired. The authors provide data from two cell lines and numerous clinical samples across three tumor types to demonstrate their methods superior performance as compared to standard ChIP-seq methods. Overall, they demonstrate a clear performance improvement over standard methods, however, additional information describing their experiments needs to be added to the manuscript.

We thank the reviewer for the constructive comments and suggestions, and we are delighted to see the reviewer appreciates our work. We now provided all additional information streams on the experiments, as was requested. With this, we sincerely hope the reviewer finds our manuscript sufficiently improved.

(1) In the section titled "DSG improves cross-linking in cell lines of transcription factors" the authors need to report the number of cells used for the ChIP-seq experiments.

We have now included cell number information in the manuscript (methods section). We would like to highlight that between conditions (DSG+ formaldehyde versus formaldehyde alone), the number of cells used in the experiments were identical.

(2) In the section titled "DSG improves cross-linking in tumor specimens of transcription factors" the authors discuss the method's applicability to profile TFs from reduced amount of frozen tissue, but they don't discuss the amount of starting tissue used for the AR/ER, FOXA1 and H3K27ac profiles shown in figures 3 and 4 or indeed any of the tissue samples. The authors should detail the starting number of sections and an estimate of the cross-section size. Also, if available, the authors should report the amount of DNA in the soluble chromatin that was used for the immuno-precipitations.

We thank the reviewer for drawing this to our attention. Now, for each of the tissue samples used, we have included details on tissue surface area, number of slices used, and slice thickness (Supplementary Table S1). Sadly, information on the amount of DNA used in the immuno-precipitation cannot be retrieved anymore at this point. That sad, since tumors are intrinsically heterogeneous and contain other cell types apart from only tumor cells as well, the total amount of DNA per se is not necessarily informative. In order to account for tumor cell

heterogeneity within our analyses, we setup the tissue ChIP-seq analyses as follows: when cryo-sectioning the tissue samples, alternating slices were collected for 'formaldehyde alone' or 'formaldehyde + DSG' processing, as indicated in Figure 1. With this setup, we believe we have minimized any effect of tumor cellularity or tumor cell heterogeneity in the comparative analyses between the two fixation methods.

(3) The authors state towards the bottom of page 7 "... and no bias in peak distribution is observed for specific genomic regions (Figure 2D)." However Figure 2D shows that the percentage of promoter peaks changes from 0% in FA to 3% in DSG+FA for AR. For ER the changes are 2.7% in FA to 10.8% in DSG+FA (a 4-fold increase). This seems to indicate a bias for promoter peaks in DSG+FA data that does not significantly detract from the method but should be acknowledged and discussed.

The reviewer is absolutely right, and this omission has now been corrected. Within this analysis, the total number of peaks analyzed is very low (AR: 109 FA-unique peaks; ER: 74 FA unique peaks). With that, we are cautious and reluctant to potentially over-interpret these differential genomic distributions of such small numbers of sites. Overall, we did not observe a strong bias of ChIP-seq reads-in-peaks per peak between the two fixation methods (Supplementary Figure S3B). That said, we agree with the reviewer in that we cannot rule out any potential selectivity of genomic regions (albeit for a relatively low number of sites), and these results are now better discussed and acknowledged in the results section.

(4) In Figure 3 the authors show ChIP-seq analyses comparing AR, FOXA1 and H3K27ac in primary in prostate tissue with similar analyses in Figures 4 and 5. It is striking that virtually all the TF data using FA appears to have completely failed (many have zero peaks and/or zero FrIP scores). This is surprising since there is significantly better quality data that has been published from these tumor types for these TFs by multiple groups (as the authors reference in their paper). There needs to be an explanation for this apparent discordance with the literature results.

This is an excellent point. As the reviewer correctly states, other reports (including multiple manuscripts from our own lab) have described ChIP-seq profiles for all the factors that were also described in this manuscript. With that, it may be surprising to see that for all the samples we analyzed in this report, the 'conventional method' which only uses formaldehyde performed quite poorly. We would like to highlight that for this report, we included practically every single sample that we analysed. For the vast majority of other publications, definitively including our own, ChIP-QPCR enrichment was used as a selection criterium to identify samples that would be eligible for sequencing. Subsequently, those samples with a low number of peaks of a low FRiP score, were typically removed from the dataset prior to analysis. With that, a rigorous sample pre-selection was

made, exclusively analyzing samples that pass specific pre-defined quality criteria. As we now included practically all samples processed, such pre-selection was not performed, explaining the low success-rate of the conventional procedure.

(5) In several places in the manuscript the authors state that there is no effect of double fixation on histone mark ChIP-seq (e.g. near the top of page 9: "The results were in line with H3K27ac results in cell lines and tumors, i.e. no consistent significant differences were observed in terms of number of peaks detected.") However, Figure 4B and 5B show consistent, large increases in the number of H3K27ac peaks identified with DSG+FA compared to FA. This increase should be acknowledged and there should be some effort to show these additional peaks are likely real (overlap with H3K27ac in cell line data, sequence conservation, etc.)

The reviewer raises an excellent point that was also brought up by the other reviewer. The reviewer is absolutely right, and statements regarding comparable profiles for histone marks between 'formaldehyde only' versus 'formaldehyde + DSG' has been adjusted accordingly. To address the question of whether the differential peaks between 'formaldehyde only' versus 'formaldehyde + DSG' are real and truly differential, we have performed the following analyses:

- 1. For all peak subsets in Figures 3B, 4B, 5B and S3A, we now incorporated the raw data visualization coverage plots in Supplementary Figures S11, S12, S13. For each of the three peaksets (shared, formaldehyde-unique, DSG+ formaldehyde-unique), signal was in fact observed for all of those locations**
- 2. For H3K27ac ChIP-seq in tumor samples, other independent datasets have been published in endometrial cancer (Droog et al., 2017 PNAS), prostate cancer (Kron et al 2017 Nature Genetics) and breast cancer (Patten et al., 2018 Nature Medicine). For each of the H3K27ac peak subsets we identified for any of the tumors, we tested whether read coverage was observed in the publicly available datasets (Supplementary Figures S15, S16, S17). For each of the peak subsets (shared, formaldehyde-unique, DSG+ formaldehyde-unique), clear coverage was found in all the other datasets tested. Please note that consistently the 'shared between formaldehyde and DSG+formaldehyde' regions consistently showed the highest coverage for all samples tested.**
- 3. The histone modification ChIP-seq data were now further confirmed using ChIP-QCPR (Supplementary Figure S14). Again, for each of the subsets of peaks, enrichment was observed for all samples tested, which was consistently enriched over IgG negative control.**

Cumulatively, these controls show that rather than unique peaks, these 'DSG + formaldehyde unique' and 'formaldehyde unique' peaks should be considered as false-negative regions that were not picked up with the peak calling algorithm that was used. This is now explicitly mentioned in the results section, and all analyses mentioned are now included in the supplementary figures.

(6) Lastly, some considerations should be made regarding chromatin fragmentation after extensive fixation. It is expected that there will be an increased proportion of high DNA sizes in the soluble chromatin after the DSG/FA protocol compared to FA alone. The high sizes will reduce the amount of DNA that can be sequenced after the IP. The authors should discuss this point and show results for the fragment size distribution comparing both DSG/FA and standard FA.

To address this point, we have included the following analyses in the manuscript:

- 1. For MCF-7 and LNCaP cells, we now provide agarose gels of the sonicated total cell lysates (Supplementary Figure S1A). Even though slightly larger fragments were found for FA-fixed MCF-7 cells, overall no major impact was seen on fragment size.**
- 2. We extensively re-analyzed the bioanalyser tracks for a number of tissue ChIP samples, prior to library preparation. Two major fragment ranges were observed: one within the fragment range size optimal for Illumina seq (~200-500bp), and one of substantial larger fragment size. Between 'formaldehyde only' and 'DSG+formaldehyde' samples, no major differences were observed in fragment size distribution. For each tumor type, one example bioanalyser plot has now been provided in Supplementary Figure S1B.**

December 18, 2018

RE: Life Science Alliance Manuscript #LSA-2018-00115-TR

Prof. Wilbert Zwart
Netherlands Cancer Institute
dept of Oncogenomics
PLesmanlaan 121
Amsterdam, NH 1066CX
Netherlands

Dear Dr. Zwart,

Thank you for submitting your revised manuscript entitled "Optimized ChIP-seq procedure facilitates transcription factor profiling in human tumors." As you will see, the reviewers appreciate the introduced changes and we would thus be happy to publish your paper in Life Science Alliance pending final revisions necessary to meet our formatting guidelines:

Please provide the supplementary figures as individual files and add callouts in the manuscript text for SFig1A (only SFig1B currently called out) and for SFig18. Please link your ORCID iD to your profile in the submission system, you should have received an email with instructions on how to do so.

A. FINAL FILES:

-- High-resolution figure, supplementary figure and video files uploaded as individual files: See our detailed guidelines for preparing your production-ready images, <http://life-science-alliance.org/authorguide>

B. MANUSCRIPT ORGANIZATION AND FORMATTING:

Full guidelines are available on our Instructions for Authors page, <http://life-science-alliance.org/authorguide>

Thank you for your attention to these final processing requirements.

Sincerely,

Reviewer #2 (Comments to the Authors (Required)):

All the concerns have been addressed in the revised version, I would accept the manuscript for publication in LSA.

Reviewer #3 (Comments to the Authors (Required)):

The paper entitled "Optimized ChIP-seq method facilitates transcription factor profiling in human tumors" by Singh et al. describes a technical improvement to increase ChIP-seq signal with particular application to transcription factors in frozen tissues. Improvements to the quality of ChIP-seq data from clinical samples and obtaining more data from limited amounts of clinical tissue are both important areas where advances are highly desired.

The authors have extensively addressed all the major points raised by the reviewers in the revised version of the manuscript. The paper is sufficiently improved to warrant publication in its current state.

December 18, 2018

RE: Life Science Alliance Manuscript #LSA-2018-00115-TRR

Prof. Wilbert Zwart
Netherlands Cancer Institute
dept of Oncogenomics
PLesmanlaan 121
Amsterdam, NH 1066CX
Netherlands

Dear Dr. Zwart,

Thank you for submitting your Methods entitled "Optimized ChIP-seq procedure facilitates transcription factor profiling in human tumors.". It is a pleasure to let you know that your manuscript is now accepted for publication in Life Science Alliance. Congratulations on this interesting work.

DISTRIBUTION OF MATERIALS:

Again, congratulations on a very nice paper. I hope you found the review process to be constructive and are pleased with how the manuscript was handled editorially. We look forward to future exciting submissions from your lab.

Sincerely,
